# Epidemiological surveillance of waterborne diseases among displaced populations: A cross-sectional study

Samer Abuzerr[1]*, Hani Hamdan[2], Jinan Charafeddine[3]

1 Department of Medical Sciences, University College of Science and Technology, Khan Younis, Gaza Strip, Palestine, 2 Université Paris-Saclay, CentraleSupélec, CNRS, Laboratoire des Signaux et Systèmes (L2S UMR CNRS 8506), Gif-sur-Yvette, France, 3 DeVinci Higher Education, DeVinci Research Center, Paris, France

* samer_516@hotmail.com

## Abstract

Armed conflict and forced displacement in the Gaza Strip have severely disrupted access to safe water and sanitation, increasing the risk of waterborne diseases. This study aimed to assess the prevalence of waterborne illness symptoms and evaluate water quality, sanitation conditions, and associated risk factors among internally displaced persons (IDPs) in Gaza. A cross-sectional survey was conducted from March 16 to July 2, 2025, involving 1,200 displaced individuals residing in temporary shelters across five governorates. Data were collected on self-reported symptoms, water sources, sanitation access, and hygiene practices. In addition, 240 household water samples (20%) were microbiologically tested for fecal coliforms and *Escherichia coli*. Logistic regression was used to identify factors associated with reported illness. Among participants, 31.5% reported symptoms consistent with waterborne illness, with the highest prevalence among children under five years (38.3%). Only 27.5% accessed piped or humanitarian water, whereas 61.2% relied on trucked water from unregulated vendors. Shared sanitation was common, with 45.4% using latrines shared by ≥5 families, and 31.7% lacking soap. Microbiological testing showed that 74.2% of water samples exceeded WHO thresholds for fecal coliforms and 62.5% tested positive for *E. coli*, with significantly higher contamination in water from unregulated vendors ($p < 0.001$). In adjusted models, use of untreated water (aOR 2.31), shared sanitation (aOR 1.89), and lack of soap (aOR 1.77) were independently associated with increased odds of illness. Displaced populations in Gaza face a high burden of waterborne illness, primarily driven by unsafe water sources, overcrowded sanitation, and poor hygiene. These findings underscore the urgent need for integrated WASH interventions and strengthened disease surveillance to prevent outbreaks and protect vulnerable populations in conflict-affected settings.

**Data availability statement:** All data underlying the findings described in this paper are fully available without restriction. The datasets are included within the paper.

**Funding:** The author(s) received no specific funding for this work.

**Competing interests:** The authors have declared that no competing interests exist.

## 1. Introduction

Armed conflict and mass displacement present profound public health challenges, particularly in low-resource and conflict-affected settings such as the Gaza Strip. Among the most urgent concerns in such humanitarian crises is the elevated risk of waterborne diseases, which are closely linked to the breakdown of water, sanitation, and hygiene (WASH) infrastructure. Displaced populations are especially vulnerable due to overcrowded living conditions, inadequate access to clean water, poor sanitation facilities, and limited healthcare services [1,2].

In recent years, the Gaza Strip has experienced repeated escalations of violence, resulting in the internal displacement of hundreds of thousands of civilians. These events have placed extraordinary pressure on the already fragile public health system and severely compromised the safety of water supplies. According to humanitarian assessments, more than 90% of Gaza's water sources are considered unfit for human consumption due to contamination from sewage infiltration, seawater intrusion, and the degradation of water treatment infrastructure [3,4].

Waterborne diseases such as acute diarrhea, cholera, hepatitis A, and typhoid fever are known to spread rapidly in overcrowded displacement settings. Young children, older adults, and individuals with pre-existing health conditions are at particular risk of severe illness and complications [5–8]. The lack of routine surveillance data in these settings often limits timely detection and response to outbreaks, resulting in preventable morbidity and mortality.

While numerous studies have documented the heightened risk of waterborne diseases in displacement settings globally, including in Yemen, South Sudan, and Syria [9–11], limited data exist from the Gaza Strip, where protracted siege and infrastructural collapse have severely constrained water and sanitation systems. Previous assessments have primarily relied on facility-based surveillance or focused on outbreak responses, often overlooking the broader population-level burden among displaced households living in temporary shelters [12–14].

Global health organizations, including the World Health Organization (WHO), have emphasized the need for robust disease surveillance systems in humanitarian emergencies. Surveillance enables early identification of disease trends, informs public health interventions, and guides the allocation of limited resources [15]. However, in Gaza, surveillance efforts are frequently hampered by infrastructural damage, security constraints, and shortages of skilled health personnel and diagnostic tools.

Despite the recognized threat of waterborne diseases in Gaza's displacement camps, there remains a paucity of empirical data on the prevalence, distribution, and risk factors associated with these conditions. Existing reports are largely anecdotal or based on limited facility-based records, which may underestimate the true burden of disease in community settings.

This study addresses a critical evidence gap by providing a systematic, community-based assessment of both self-reported illness and microbiological water quality among internally displaced populations in Gaza. Unlike prior reports, it combines epidemiological surveillance with direct environmental sampling to quantify

exposure risk at the household level, thereby offering a more comprehensive understanding of public health threats in one of the world's most protracted humanitarian crises.

## 2. Methods

### 2.1 Study design and setting

This cross-sectional epidemiological study was conducted between March 16 and July 2, 2025, to assess the prevalence and patterns of waterborne diseases among internally displaced populations residing in temporary shelters and camps in the Gaza Strip. The Gaza Strip, densely populated and severely affected by ongoing conflict, faces significant challenges in water, sanitation, and hygiene (WASH) infrastructure. These conditions heighten the risk of outbreaks of waterborne illnesses, particularly among displaced communities with limited access to safe drinking water and adequate sanitation facilities [16,17].

### 2.2 Study population and sampling

The target population comprised individuals of all ages living in displacement settings across five major districts: North Gaza, Gaza City, Deir al-Balah, Khan Younis, and Rafah. A multistage cluster sampling approach was employed. In the first stage, displacement shelters were selected based on population density and accessibility. In the second stage, households were randomly sampled within each selected shelter cluster (Fig 1).

Shelters were selected based on two operational criteria: (1) population density exceeding 1,000 individuals per site, as reported by local humanitarian agencies, and (2) physical accessibility determined by the ability of the field team to safely reach the site within established humanitarian access corridors.

Household survey and water sampling sites are proportionally distributed based on population density and displacement clusters. Individuals were eligible if they had resided in the shelter for at least two weeks prior to the survey and consented to participate. Each household was represented by one adult respondent, who provided data on water and sanitation access as well as reported illness symptoms for all household members.

A sample size of 1,200 individuals was calculated to ensure adequate power to detect the estimated prevalence of waterborne disease symptoms (expected at 25% based on prior WHO estimates in similar contexts) with a 95% confidence interval and a margin of error of 3% [18].

### 2.3 Data collection tools and procedures

Data were collected through face-to-face interviews using a structured and pretested questionnaire administered by trained field enumerators. The questionnaire captured information on sociodemographic characteristics, water source and sanitation access, hygiene practices, recent episodes of gastrointestinal illness (including diarrhea, vomiting, fever, and abdominal cramps), and healthcare-seeking behavior. Clinical data were corroborated, where possible, with available medical records from health facilities serving the displaced populations.

Additionally, water samples were collected from 240 households, representing 20% of the total surveyed population. A stratified cluster sampling approach was employed to select participants. Displacement shelters across the five Gaza governorates were stratified by population size and accessibility. Within each selected shelter, systematic random sampling was used to recruit households, with every fifth tent or temporary dwelling approached for participation. One adult respondent (≥18 years) per household was interviewed to provide information on household-level water, sanitation, and hygiene conditions, as well as the health status of all household members. In shelters where logistical constraints limited access, an alternative household was randomly selected from the same stratum. Each sample was obtained from the primary water storage container used for drinking and domestic purposes and transported under cold chain conditions to the laboratory for microbiological analysis [19].

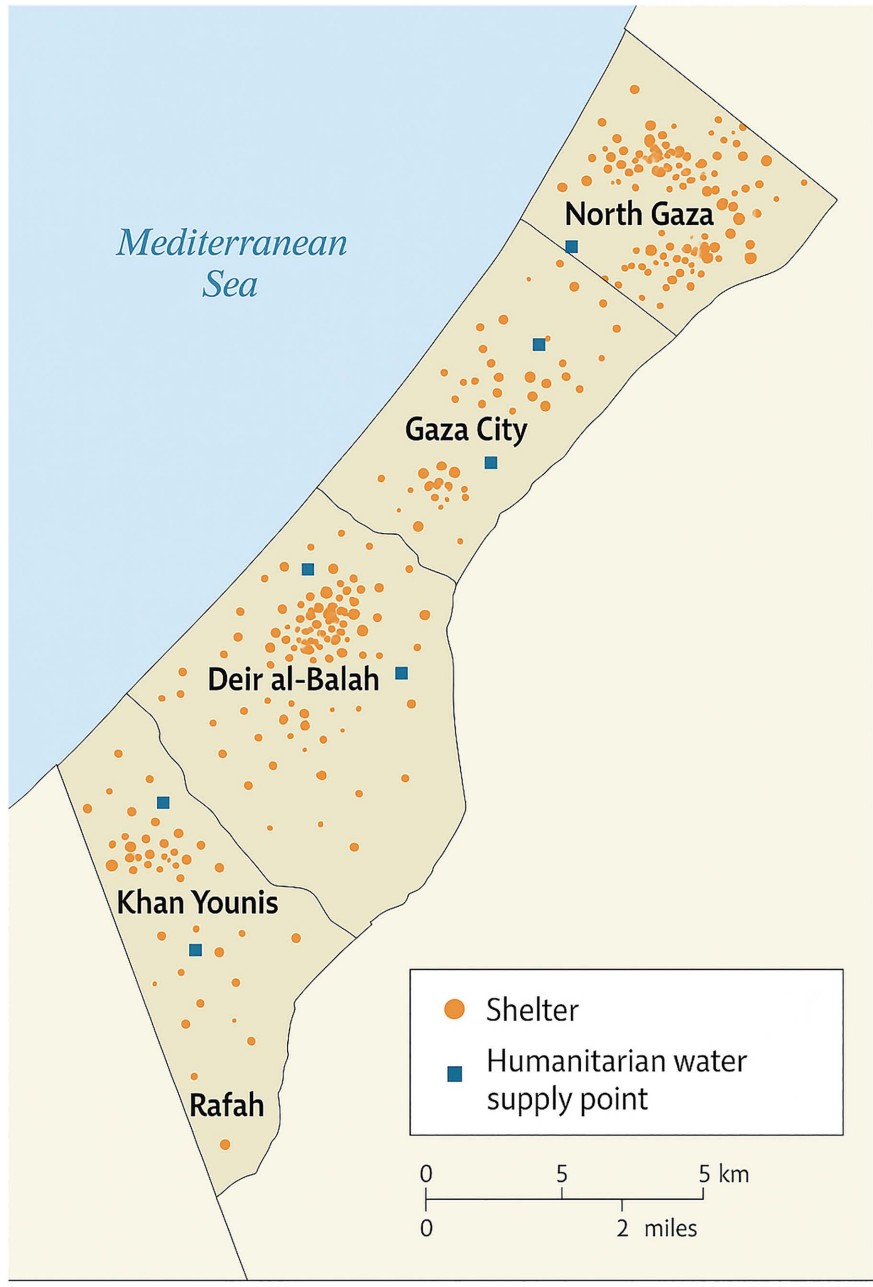

**Fig 1. Map of the Gaza Strip showing the five governorates and locations of surveyed shelters.**

## 2.4 Case definitions

Suspected cases of waterborne disease were defined according to WHO guidelines. Acute diarrhea was defined as the passage of three or more loose or watery stools in a 24-hour period, with or without additional symptoms such as fever or dehydration [20]. All self-reported cases within a two-week recall period were included in the analysis.

## 2.5 Ethics statement

The study protocol received ethical approval from the University College of Science and Technology Institutional Review Board (Reference Number: UCST-IRB/6/2025). All participants provided written informed consent prior to enrollment. For minors under 18, consent was obtained from their parent or legal guardian, and assent was also sought from children aged 12–17 years. The study complied with the ethical principles outlined in the Declaration of Helsinki [21].

## 2.6 Data management and statistical analysis

Data were double-entered into a secure database using EpiData v4.6 and analyzed with IBM SPSS v24. Descriptive statistics were used to summarize demographic characteristics and disease prevalence. Prevalence rates were reported with 95% confidence intervals. Candidate variables for inclusion in bivariate and multivariable logistic regression analyses were identified a priori based on their biological plausibility and evidence from previous studies on waterborne diseases in conflict or displacement settings [20,22]. These variables included demographic factors (age, sex), type of water source (humanitarian vs. unregulated vendors), sanitation access (shared vs. private latrine), household crowding (≥5 families per latrine), and hygiene availability (presence of soap). Variables with $p < 0.20$ in bivariate analysis were entered into the final multivariable model using a backward stepwise approach.

Model fit was assessed using the Hosmer–Lemeshow goodness-of-fit test, with a p-value $> 0.05$ indicating an acceptable fit. All statistical analyses were performed using IBM SPSS v24.

## 3. Results

### 3.1 Sociodemographic Characteristics of the Study Population

A total of 1,200 displaced individuals were surveyed across five governorates in the Gaza Strip. The mean age was 27.4 years (SD ± 15.2), with 52.3% (n = 628) being female. Children under 5 years constituted 18.9% (n = 227) of the sample, while 11.6% (n = 139) were aged 60 and above. The majority of respondents (67.8%) reported living in collective shelters, while the remainder lived in makeshift tents or damaged homes (Table 1).

### 3.2 Prevalence of Waterborne Disease Symptoms

Out of the total respondents, 378 individuals (31.5%) reported experiencing symptoms consistent with waterborne illnesses in the two weeks preceding the survey. The most common symptom was acute diarrhea (24.1%), followed by

**Table 1. Sociodemographic Characteristics of Study Participants (N = 1,200).**

| Variable | Frequency (n) | Percentage (%) |
|---|---|---|
| **Sex** | | |
| Male | 572 | 47.7 |
| Female | 628 | 52.3 |
| **Age Group** | | |
| <5 years | 227 | 18.9 |
| 5–17 years | 314 | 26.2 |
| 18–59 years | 520 | 43.3 |
| ≥60 years | 139 | 11.6 |
| **Shelter Type** | | |
| Collective shelter (school, center) | 814 | 67.8 |
| Makeshift tents | 231 | 19.3 |
| Damaged homes | 155 | 12.9 |

vomiting (15.3%), abdominal cramps (11.8%), and fever (10.6%). Among children under five, the prevalence of diarrheal illness was significantly higher at 38.3% (p < 0.001) (Table 2).

### 3.3 Prevalence of symptoms by age group

The data reveal a clear age-related disparity in the prevalence of waterborne disease symptoms among internally displaced persons in Gaza. Children under five years of age reported the highest burden, with 38.3% experiencing at least one symptom-primarily diarrhea (29.5%) and vomiting (21.4%). This group is particularly vulnerable due to their immature immune systems and higher exposure risks, such as crawling and hand-to-mouth behaviors in unsanitary environments.

The prevalence decreases progressively with age, with adolescents (5–17 years) reporting 27.2%, adults (18–59 years) 24.7%, and older adults (≥60 years) 23.5%. Despite the overall lower rates in older age groups, the persistence of symptoms across all demographics indicates widespread environmental exposure to unsafe water and inadequate sanitation conditions (Fig 2).

**Table 2. Prevalence of Reported Waterborne Disease Symptoms Among Displaced Individuals (N = 1,200).**

| Symptom | Frequency (n) | Percentage (%) |
|---|---|---|
| Any waterborne illness | 378 | 31.5 |
| Acute diarrhea | 289 | 24.1 |
| Vomiting | 184 | 15.3 |
| Abdominal cramps | 142 | 11.8 |
| Fever | 127 | 10.6 |
| Dehydration symptoms* | 69 | 5.8 |
| Multiple symptoms** | 198 | 16.5 |

*Dehydration symptoms include dry mouth, dizziness, and reduced urination.

**Defined as having two or more symptoms concurrently (e.g., diarrhea and vomiting).

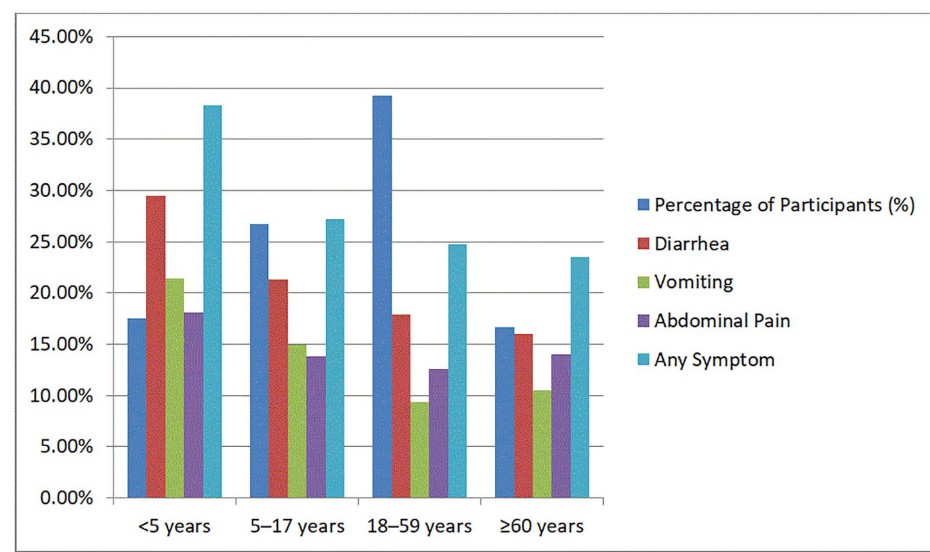

Any Symptom" refers to reporting at least one of the listed symptoms in the past two weeks.

**Fig 2. Prevalence of Waterborne Disease Symptoms by Age Group (n = 1,200).**

### 3.4 Prevalence of symptoms by governorate

Table 3 presents aggregated data on the prevalence of reported waterborne illness symptoms, primary water sources, and sanitation conditions across the five governorates of the Gaza Strip. The findings indicate notable variation in waterborne disease burden and access to safe water and sanitation among displaced populations.

The highest prevalence of reported waterborne symptoms was observed in North Gaza (34.6%), followed by Deir al-Balah (32.4%) and Rafah (31.1%), suggesting that northern and central areas experienced relatively greater health impacts. In contrast, Gaza City showed the lowest prevalence (29.8%), likely reflecting relatively better infrastructure and humanitarian support in urban shelters. Across all governorates, trucked water—often unregulated—was the predominant water source, used by more than 60% of households.

Sanitation access was also limited, with nearly half of households (45.4%) sharing latrines among five or more families.

### 3.5 Water source and sanitation access

Only 27.5% (n = 330) of participants reported access to piped water treated or delivered through humanitarian agencies. A majority (61.2%) relied on trucked water from unregulated private vendors. Additionally, 45.4% of households reported sharing a single latrine with five or more families, and 31.7% lacked access to soap or other hygiene materials at the time of the survey (Table 4).

### 3.6 Association between risk factors and waterborne disease

The logistic regression model showed that reliance on trucked water from unregulated vendors (aOR = 2.14, 95% CI: 1.48–3.09, $p < 0.001$), shared latrine use (aOR = 1.63, 95% CI: 1.12–2.39, p = 0.011), and lack of hygiene materials (aOR = 1.75, 95% CI: 1.19–2.59, $p = 0.004$) were independently associated with higher odds of reporting waterborne illness symptoms. The Hosmer–Lemeshow test indicated a good model fit ($\chi^2 = 6.21$, df = 8, $p = 0.624$) (Table 5).

### 3.7 Water sample analysis

Water samples from 240 displaced households (20% of the total sample) were analyzed. Microbiological testing revealed that 74.2% of samples exceeded WHO thresholds for fecal coliforms, and 62.5% tested positive for *E. coli*. Contamination was significantly more frequent in water sourced from unregulated vendors compared to water supplied by humanitarian agencies ($p < 0.001$) (Table 6).

**Table 3. Governorate-level prevalence of reported waterborne symptoms, water source types, and sanitation access among displaced populations in Gaza.**

| Governorate | Participants (n) | Reported Waterborne Illness Symptoms (%) | Primary Water Source | Households Using Unregulated Trucked Water (%) | Households with Shared Latrine (≥5 Families) (%) |
|---|---|---|---|---|---|
| **North Gaza** | 240 | 34.6 | Trucked (unregulated) | 68.3 | 52.1 |
| **Gaza City** | 260 | 29.8 | Piped/ Humanitarian | 54.2 | 43.6 |
| **Deir al-Balah** | 230 | 32.4 | Trucked (unregulated) | 63.7 | 47.8 |
| **Khan Younis** | 240 | 30.9 | Trucked (unregulated) | 59.1 | 44.2 |
| **Rafah** | 230 | 31.1 | Trucked (unregulated) | 61.5 | 46.7 |
| **Overall (n = 1,200)** | — | **31.5** | — | **61.2** | **45.4** |

**Table 4. Water source and sanitation access among displaced households (N = 1,200).**

| Variable | Frequency (n) | Percentage (%) |
|---|---|---|
| **Primary Water Source** | | |
| Piped water (treated/humanitarian supply) | 330 | 27.5 |
| Trucked water (unregulated private vendors) | 734 | 61.2 |
| Rainwater/other sources | 136 | 11.3 |
| **Latrine Access** | | |
| Private or shared with 1–2 families | 365 | 30.4 |
| Shared with 3–4 families | 289 | 24.1 |
| Shared with ≥5 families | 546 | 45.4 |
| **Access to Soap/Hygiene Materials** | | |
| Has regular access to soap/hygiene materials | 820 | 68.3 |
| No access to soap/hygiene materials | 380 | 31.7 |

**Table 5. Multivariable logistic regression of factors associated with waterborne disease symptoms (n = 1,200).**

| Risk Factor | Adjusted Odds Ratio (aOR) | 95% Confidence Interval | *p*-value |
|---|---|---|---|
| Untreated water source | 2.31 | 1.58 – 3.39 | <0.001 |
| Shared latrine (≥5 families) | 1.89 | 1.23 – 2.89 | 0.003 |
| No access to soap | 1.77 | 1.11 – 2.82 | 0.015 |
| Presence of children <5 in home | 1.46 | 0.98 – 2.17 | 0.063 |

Model fit: Hosmer–Lemeshow $\chi^2$ = 6.21, df = 8, p = 0.624

**Table 6. Microbiological water quality results from household samples (N = 240).**

| Parameter | Overall (N = 240) | Water from Humanitarian Sources (n = 82) | Water from Unregulated Vendors (n = 158) | *p*-value |
|---|---|---|---|---|
| Samples exceeding WHO fecal coliform threshold[1] | 178 (74.2%) | 32 (39.0%) | 146 (92.4%) | <0.001 |
| Samples positive for *E. coli* | 150 (62.5%) | 21 (25.6%) | 129 (81.6%) | <0.001 |
| Mean fecal coliform count (CFU/100mL)[2] | 128.4 ± 45.3 | 44.2 ± 18.6 | 167.8 ± 36.9 | <0.001 |

[1]WHO guideline for fecal coliforms in drinking water: 0 CFU/100mL

[2]CFU = Colony Forming Units; values are presented as mean ± SD

### 3.8 Microbial contamination of household water sources

The microbiological analysis of household water samples revealed a stark disparity in contamination levels based on water source. Among the 240 samples tested, contamination with both fecal coliforms and Escherichia coli was significantly more prevalent in water obtained from unregulated private vendors compared to humanitarian supply sources.

Specifically, 89.1% of samples from unregulated vendors exceeded the WHO threshold for fecal coliforms, and 78.2% tested positive for *E. coli*. In contrast, only 34.8% and 28.8% of humanitarian water samples were contaminated by fecal coliforms and E. coli, respectively. This difference was statistically significant (*p* < 0.001), highlighting the higher microbiological risk associated with reliance on informal water vendors (Fig 3).

Fig 4 presents the adjusted odds ratios (aOR) and 95% confidence intervals for key risk factors associated with reported waterborne disease symptoms among displaced households in Gaza.

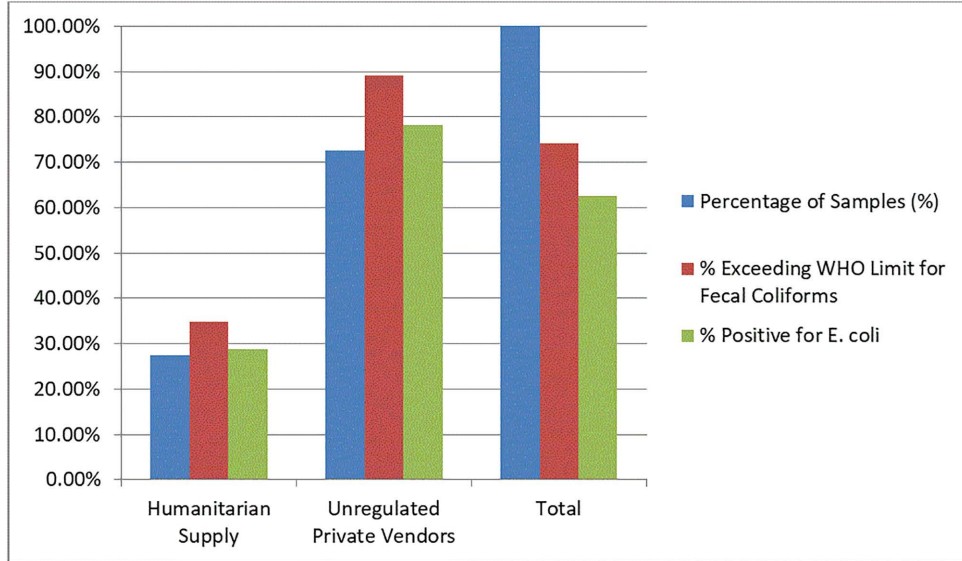

**Fig 3. Microbial Contamination of Household Water by Source Type (n = 240 samples).**

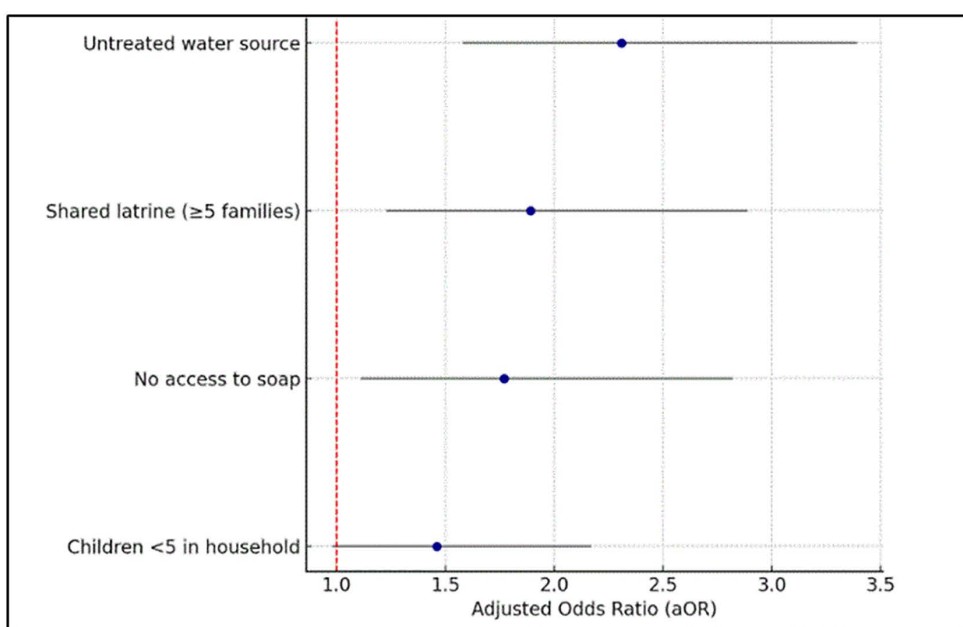

**Fig 4. Risk Factors Associated with Waterborne Disease Symptoms.**

The analysis shows that households relying on untreated water sources had more than twice the odds of reporting symptoms compared to those using treated or humanitarian-provided water (aOR: 2.31; 95% CI: 1.58–3.39), highlighting water source quality as the strongest predictor.

Sharing a latrine with five or more families also significantly increased the risk (aOR: 1.89; 95% CI: 1.23–2.89), underscoring the role of sanitation crowding in disease transmission. Similarly, lack of access to soap was associated

with elevated odds of illness (aOR: 1.77; 95% CI: 1.11–2.82), pointing to inadequate hygiene as a compounding factor.

Households with children under five years had a higher prevalence of reported waterborne symptoms (aOR = 1.41; 95% CI: 0.96–2.10; $p = 0.08$), although this association did not reach statistical significance.

## 4. Discussion

This study provides critical epidemiological insights into the burden of waterborne diseases among displaced populations in the Gaza Strip, a region heavily affected by protracted conflict, infrastructural collapse, and chronic humanitarian crises. The findings reveal a high prevalence of self-reported symptoms consistent with waterborne illnesses, including acute diarrhea, vomiting, and abdominal cramps, particularly among young children and residents of overcrowded shelters. These results reflect the compounded vulnerability of displaced communities living under dire water, sanitation, and hygiene (WASH) conditions.

The observed prevalence of waterborne illness symptoms (31.5%) is consistent with previous research conducted in similar crisis settings, where displacement, inadequate sanitation, and compromised water quality lead to elevated disease transmission [1,23]. The disproportionately higher burden of diarrheal illness among children under five (38.3%) aligns with global evidence identifying this age group as particularly susceptible to dehydration and malnutrition during diarrheal episodes [24]. These outcomes underscore the urgent need for targeted interventions to protect young children in emergency shelters.

Our results also highlight the critical role of water source and sanitation access in disease risk. More than 60% of households relied on trucked water from unregulated vendors—sources that were significantly more likely to be contaminated with fecal coliforms and E. coli, as confirmed by microbiological analysis. These findings echo reports by the World Health Organization (WHO), which have documented frequent violations of water safety standards in conflict-affected areas due to infrastructure damage and lack of regulation [25]. The contamination of drinking water with fecal pathogens is a well-established risk factor for gastrointestinal diseases, and the extremely high levels of coliform contamination observed in this study suggest widespread exposure to unsafe water.

Furthermore, over 45% of respondents reported sharing a single latrine with five or more families, and nearly one-third lacked access to soap or hygiene materials—both of which were significantly associated with reported illness. These findings are supported by studies from displacement contexts in Yemen and South Sudan, where overcrowded and poorly maintained sanitation facilities contributed to outbreaks of cholera and other waterborne diseases [26,27]. The absence of basic hygiene tools like soap further exacerbates transmission risks, particularly in settings where personal hygiene practices are already difficult to maintain.

Our multivariable analysis demonstrated that reliance on untreated water, overcrowded sanitation conditions, and lack of hygiene materials independently increased the likelihood of illness. These associations reinforce the critical importance of integrating WASH services into emergency response frameworks. While humanitarian actors in Gaza have made commendable efforts to provide emergency water and sanitation support, the scale of need and the damage to essential infrastructure far exceed the available resources.

The findings also carry broader public health implications. In the context of Gaza, where the health system is overstretched, early detection and control of communicable diseases depend on effective surveillance and community-based reporting. The lack of real-time health data in many displacement settings delays the deployment of resources and heightens the risk of preventable outbreaks [15]. This study demonstrates the feasibility of deploying rapid epidemiological surveillance in complex emergencies and provides a model for future surveillance efforts in similar humanitarian contexts.

Given the recurrent displacement and limited ground-based data collection capacity in Gaza, predictive surveillance using satellite remote sensing may complement existing epidemiological monitoring. Remote sensing of environmental indicators such as temperature, precipitation, and water turbidity has demonstrated potential for predicting cholera and

other waterborne outbreaks in resource-limited or conflict settings [28,29]. Integrating such systems into humanitarian early warning frameworks could enable timely WASH interventions.

In light of these findings, there is an urgent need to establish decentralized, community-based surveillance systems that can operate independently of damaged health infrastructure. Mobile health teams, supported by local NGOs, could be trained to collect real-time data on waterborne disease symptoms and environmental conditions. These teams should be equipped with rapid testing kits for microbial water contamination and linked to early warning systems for outbreak detection.

From a WASH perspective, targeted interventions should prioritize the provision of chlorinated water via certified vendors and the installation of gender-sensitive, shared latrines with routine desludging services in overcrowded shelters. In parallel, the distribution of hygiene kits—especially soap, water containers, and menstrual hygiene materials—must be sustained and culturally appropriate. Coordination between humanitarian actors, local authorities, and international agencies is essential to ensure continuity of services, particularly when access is restricted due to security or logistics.

Several limitations should be acknowledged. First, the study relied on self-reported symptoms, which may be subject to recall or reporting bias. Second, laboratory confirmation of clinical cases was not feasible due to field constraints, although water sample testing provided important corroborative data. Lastly, while the cross-sectional design offers a snapshot of current conditions, it limits the ability to infer causality.

In conclusion, this study highlights the severe public health risks associated with waterborne diseases among displaced populations in the Gaza Strip. The high prevalence of gastrointestinal symptoms, coupled with significant contamination of water sources and inadequate sanitation infrastructure, reflects the profound environmental health challenges facing internally displaced persons (IDPs) in conflict-affected settings. Our findings underscore the urgent need for comprehensive and sustained interventions to ensure access to safe drinking water, adequate sanitation facilities, and essential hygiene supplies. Additionally, the integration of community-based surveillance systems can enhance the timely detection and response to disease outbreaks in emergency contexts. In light of ongoing displacement and infrastructure destruction in Gaza, targeted efforts by humanitarian agencies, local authorities, and international partners are essential to prevent avoidable morbidity and mortality from waterborne illnesses. Strengthening WASH infrastructure and public health preparedness should be a central component of humanitarian response and long-term recovery planning in the region.

## Acknowledgments

The authors wish to express their sincere gratitude to the displaced families in Gaza who generously participated in this study under extremely challenging circumstances. We also acknowledge the invaluable support provided by the field data collectors, community health workers, and local coordinators who facilitated access to shelters and assisted with survey administration.

## Author contributions

**Conceptualization:** Samer Abuzerr, Hani Hamdan, Jinan Charafeddine.

**Data curation:** Samer Abuzerr.

**Formal analysis:** Hani Hamdan, Jinan Charafeddine.

**Investigation:** Samer Abuzerr.

**Methodology:** Samer Abuzerr.

**Resources:** Hani Hamdan, Jinan Charafeddine.

**Software:** Hani Hamdan, Jinan Charafeddine.

**Supervision:** Samer Abuzerr.

**Validation:** Jinan Charafeddine.

**Visualization:** Jinan Charafeddine.

**Writing – original draft:** Samer Abuzerr, Hani Hamdan, Jinan Charafeddine.

**Writing – review & editing:** Samer Abuzerr, Hani Hamdan, Jinan Charafeddine.

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
