## [Decision Letter · Decision Letter 0]

16 Sep 2025

PGPH-D-25-01756

Epidemiological Surveillance of Waterborne Diseases Among Displaced Populations: A Cross-Sectional Study

Dear Dr. Abuzerr,

Thank you for submitting your manuscript to PLOS Global Public Health. After careful consideration, we feel that it has merit but does not fully meet PLOS Global Public Health’s publication criteria as it currently stands. Therefore, we invite you to submit a revised version of the manuscript that addresses the points raised during the review process.

Your manuscript has been reviewed by two reviewers, and their comments are available below. Please revise your manuscript carefully to address the points raised.

We look forward to receiving your revised manuscript.

Kind regards,

Jenna Scaramanga

Staff Editor

Journal Requirements:

2. In the online submission form, you indicated that Datasets are available from the corresponding author upon reasonable request, in accordance with data transparency and ethical guidelines.

3. Uploaded as supplementary information.

3. Please amend your detailed Financial Disclosure statement. This is published with the article. It must therefore be completed in full sentences and contain the exact wording you wish to be published.

If you did not receive any funding for this study, please simply state: “The authors received no specific funding for this work.

4. Please send a completed 'Competing Interests' statement, including any COIs declared by your co-authors. If you have no competing interests to declare, please state "The authors have declared that no competing interests exist". Otherwise please declare all competing interests beginning with the statement "I have read the journal's policy and the authors of this manuscript have the following competing interests:"

5. Please provide separate figure files in .tif or .eps format.

6. Some material included in your submission may be copyrighted. According to PLOS’s copyright policy, authors who use figures or other material (e.g., graphics, clipart, maps) from another author or copyright holder must demonstrate or obtain permission to publish this material under the Creative Commons Attribution 4.0 International (CC BY 4.0) License used by PLOS journals. Please closely review the details of PLOS’s copyright requirements here: PLOS Licenses and Copyright. If you need to request permissions from a copyright holder, you may use PLOS's Copyright Content Permission form.

Potential Copyright Issues:

Figure 1: please (a) provide a direct link to the base layer of the map (i.e., the country or region border shape) and ensure this is also included in the figure legend; and (b) provide a link to the terms of use / license information for the base layer image or shapefile. We cannot publish proprietary or copyrighted maps (e.g. Google Maps, Mapquest) and the terms of use for your map base layer must be compatible with our CC-BY 4.0 license. 

Reviewers' comments:

Reviewer's Responses to Questions

**Comments to the Author**

1. Does this manuscript meet PLOS Global Public Health’s publication criteria?

Reviewer #1: Yes

Reviewer #2: Yes

2. Has the statistical analysis been performed appropriately and rigorously?

Reviewer #1: Yes

Reviewer #2: Yes

3. Have the authors made all data underlying the findings in their manuscript fully available (please refer to the Data Availability Statement at the start of the manuscript PDF file)?

Reviewer #1: No

Reviewer #2: Yes

4. Is the manuscript presented in an intelligible fashion and written in standard English?

Reviewer #1: Yes

Reviewer #2: Yes

Reviewer #1: I read the manuscript, "Epidemiological Surveillance of Waterborne Diseases Among Displaced Populations: A Cross-Sectional Study," which provides information on the prevalence of waterborne diseases in Gaza in the spring and summer months of 2025 among surveyed participants. The study also looks at the associated risk factors of these illnesses by using logistic regression and confirms the presence of contaminated water sources through water sampling at from displaced households. This study provides important information on the prevalence of waterborne illnesses in Gaza among the displaced populations. I believe much of the study is conducted well, but there are a few areas for potential improvement. First, I believe the authors should more explicitly define the candidate variables included for the bivariate and logistic regression analyses, including justification of the selected parameters. Also, a detailed map showing the prevalence of the surveyed illnesses would be very interesting to see. Similarly, mapping the reliance on different water sources, latrine access, etc. would add some insights. However, I understand the sensitivity of such information and concede making the data public may be an issue. Additionally, it may be good to include the possibility of interventions for waterborne illness being made on the basis of predictive intelligence employing satellite remote sensing of environmental conditions associated with waterborne diseases, like cholera (https://doi.org/10.1038/s41598-022-22946-y). The authors mention the possibility of targeted interventions relying on real-time data, and this approach could supplement their efforts in areas of high displacement, such as Gaza, where data is limited. On a minor note, double check grammar errors in the manuscript. One that I noted was at the conclusion start "In concludsion."

Reviewer #2: In this study, the authors aim to investigate how conflict-driven displacement in Gaza affects access to safe water, sanitation, and hygiene (WASH) and how these factors contribute to the burden of waterborne diseases. By combining survey data with microbiological testing of household water samples, the authors demonstrate the link between unsafe water, inadequate sanitation, and poor hygiene with increased risk of disease. In this study authors tried to generate evidence that underscores the urgent need for integrated WASH interventions and improved disease surveillance to protect vulnerable displaced populations in conflict-affected settings.

However, the article has some limitations, which are mentioned below:

1) The selection process for participants needs to be described in greater detail.

2) Additional references are required to support key statements. For example, the claim “While numerous studies have documented the heightened risk of waterborne diseases in displacement settings globally, including in regions such as Yemen, South Sudan, and Syria, limited data exist from the Gaza Strip under conditions of ongoing siege and acute infrastructural collapse” should be backed with citations. Several other statements throughout the manuscript also lack appropriate references.

3) In the Methods section, the authors state that displacement shelters were selected based on population density and accessibility. However, no details—such as threshold values or specific methods used—are provided for this selection process.

4) The terms individuals and households appear to be used interchangeably without clarification. It should be specified whether only one individual was selected per household, and this detail should be explicitly cited in the text.

5) The line “Variables with a p-value < 0.2 in bivariate analysis were included in the multivariate model” is redundant, as it was already mentioned in the previous paragraph.

6) The study notes that the presence of children under five years showed “borderline significance.” This should be clarified, as statistically it appears to be non-significant

**Do you want your identity to be public for this peer review?** For information about this choice, including consent withdrawal, please see our Privacy Policy

Reviewer #1: No

Reviewer #2: No

---

## [Decision Letter · Decision Letter 1]

9 Nov 2025

Epidemiological Surveillance of Waterborne Diseases Among Displaced Populations: A Cross-Sectional Study

PGPH-D-25-01756R1

Dear Dr Abuzerr,

We are pleased to inform you that your manuscript 'Epidemiological Surveillance of Waterborne Diseases Among Displaced Populations: A Cross-Sectional Study' has been provisionally accepted for publication in PLOS Global Public Health.

Best regards,

Julia Robinson

Executive Editor

Reviewer Comments (if any, and for reference):

Reviewer's Responses to Questions

**Comments to the Author**

Reviewer #1: All comments have been addressed

Reviewer #2: All comments have been addressed

publication criteria?

Reviewer #1: Yes

Reviewer #2: Yes

3. Has the statistical analysis been performed appropriately and rigorously?

Reviewer #1: Yes

Reviewer #2: Yes

4. Have the authors made all data underlying the findings in their manuscript fully available (please refer to the Data Availability Statement at the start of the manuscript PDF file)?

Reviewer #1: Yes

Reviewer #2: Yes

5. Is the manuscript presented in an intelligible fashion and written in standard English?

Reviewer #1: Yes

Reviewer #2: Yes

Reviewer #1: The authors have addressed all reviewer comments, and I have no further comments to provide. Therefore, the manuscript is ready for publication in my view.

Reviewer #2: The authors have successfully addressed all the comments.

**Do you want your identity to be public for this peer review?** For information about this choice, including consent withdrawal, please see our Privacy Policy

Reviewer #1: No

Reviewer #2: No
